# LumiNet: The Bright Side of Perceptual Knowledge Distillation

## Abstract

In knowledge distillation research, feature-based methods have dominated due to their ability to effectively tap into extensive teacher models. In contrast, logit-based approaches are considered to be less adept at extracting hidden 'dark knowledge' from teachers. To bridge this gap, we present *LumiNet*, a novel knowledge-transfer algorithm designed to enhance logit-based distillation. We introduce a perception matrix that aims to recalibrate logits through adjustments based on the model's representation capability. By meticulously analyzing intra-class dynamics, *LumiNet* reconstructs more granular inter-class relationships, enabling the student model to learn a richer breadth of knowledge. Both teacher and student models are mapped onto this refined matrix, with the student's goal being to minimize representational discrepancies. Rigorous testing on benchmark datasets (CIFAR-100, ImageNet, and MSCOCO) attests to *LumiNet*'s efficacy, revealing its competitive edge over leading feature-based methods. Moreover, in exploring the realm of transfer learning, we assess how effectively the student model, trained using our method, adapts to downstream tasks. Notably, when applied to Tiny ImageNet, the transferred features exhibit remarkable performance, further underscoring LumiNet's versatility and robustness in diverse settings. With *LumiNet*, we hope to steer the research discourse towards a renewed interest in the latent capabilities of logit-based knowledge distillation.

## 1 Introduction

The advancement in deep learning models has accelerated significant increases in both complexity and performance. However, this progress brings challenges associated with computational demands and model scalability. To mitigate this, Knowledge Distillation (KD) has been proposed as an efficient strategy (Hinton et al., 2015) to transfer knowledge from a larger, intricate model (teacher) to a more compact, simpler model (student). The primary objective is to trade off performance and computational efficiency. There are two ways of KD: logit- and feature-based strategies (Romero et al., 2014; Tian et al., 2019; r21, 2019; Yim et al., 2017). The logit-based methods aim to match the output distributions of the teacher and student models (Zhang et al., 2018; Mirzadeh et al., 2020; Zhao et al., 2022). In contrast, the feature-based methods are centered around aligning the intermediate layer representations between the two models (Romero et al., 2014). In general, feature-based KD outperforms logit-based KD in performance (Zhao et al., 2022). However, feature-based KD suffers from layer misalignment (Huang & Wang, 2017; Romero et al., 2014) (reducing sample density in this space), privacy concerns (Goodfellow et al., 2014) (intermediate model layers accessible for adversarial attacks revealing training data and posing significant threats), and escalating computational requirements (Vaswani et al., 2017; Zhao et al., 2022). These issues underscore the potential merits of the logit-based KD over feature-based KD. In light of these insights, this paper seeks to amplify the efficacy of the logit-based KD method, capitalizing on its inherent strengths.

Several reasons underpin the disparity between logit- and feature-based KD. Firstly, logit-based KD tends to struggle with granularity. Feature-based methods leverage a broader spectrum of the teacher's knowledge by aligning intermediate representations, providing richer information to the student (Heo et al., 2019a; Bengio et al., 2013; Wang & Yoon, 2021). In contrast, logits provide a more condensed representation, which might not always encapsulate the entirety of the teacher's knowledge (Romero et al., 2014). Secondly, when the teacher model has particularly high confidence in its target class, it can pose challenges. Even though temperature scaling (Hinton et al.,

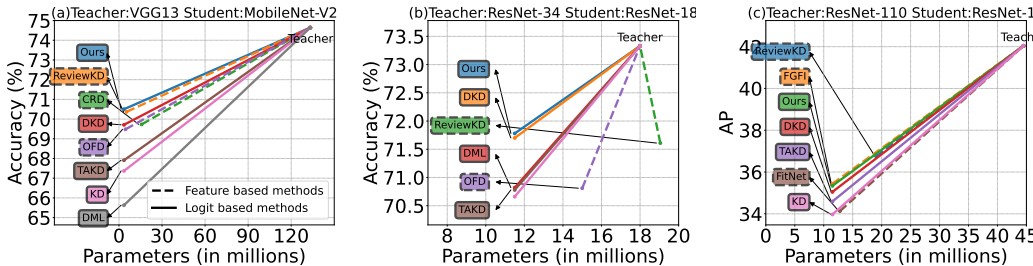

Figure 1: Performance comparison of feature-based and logit-based methods on **(a)** CIFAR-100, **(b)** ImageNet, and **(c)** MS COCO datasets. Our proposed *LumiNet*, a logit-based method, achieves high accuracy without using extra parameters.

2015) is employed to address this, determining the optimal value to achieve proper alignment in learning remains an issue (Kim et al., 2021; Chen et al., 2021a; Wang & Yoon, 2021). Thirdly, most of the logit-based methods often employ a simplistic matching criterion, which might not be robust enough to handle complex data distributions, leading to suboptimal knowledge transfer (Romero et al., 2014; Chen et al., 2021a; Wang & Yoon, 2021). Recognizing these inherent issues, we embarked on a journey to address these challenges, aspiring to elevate the performance of logit-based KD beyond that of the feature-based approach.

In response to the highlighted challenges inherent in traditional logit-based KD, we present *LumiNet*, a novel approach to knowledge distillation. *LumiNet* is inspired by human perception (Johnson, 2021): our ability to interpret objects based on context. Central to *LumiNet* is the objective of holistically reconstructing instance-level distributions. For a given batch of images spanning multiple classes, each image's logits distribution is recalibrated, considering the distributions of its batch counterparts. This reconstruction operates by normalizing intra-class logits using variance and mean. Each class's column is systematically adjusted based on its inherent variance and mean, resulting in a refined 'perception' matrix. Notably, this transformative process is applied to both the teacher and the student models. The reconstructed logits maintain consistent variance scales across the teacher and student, ensuring a harmonized knowledge transfer. However, at the instance level, we attain a fresh logit distribution due to the reconstruction, enabling the student model to harness deeper insights through the KL divergence with the teacher's output. By adopting this innovative strategy, *LumiNet* effectively addresses the challenges associated with logit-based KD: it surmounts the granularity issue by tapping into a broader knowledge spectrum, realigns the magnitude variations between teacher and student, and offers a robust matching criterion adept at handling intricate data distributions. The performance of *LumiNet* has been evaluated on three computer vision tasks: image recognition, detection for model compression, and transfer learning for feature transfer ability. Our empirical evaluations solidify *LumiNet*'s efficacy: for instance, employing ResNet8x4 as a student, we achieved a standout 77.5% accuracy and further established benchmark supremacy across tasks like CIFAR100, ImageNet, MS-COCO and TinyImageNet. Our research contributions are:

- We present *LumiNet*, a new perspective on knowledge distillation that emphasizes the reconstruction of instance-level distributions, offering a novel logit-based KD approach.
- *LumiNet* distinguishes itself with innovative features, notably its intra-class approach grounded on variance and mean. This creates a 'perception' matrix, ensuring harmonized variance between the teacher and student models, an aspect not previously addressed in the KD landscape.
- Through extensive empirical evaluations, we demonstrate that our method consistently enhances performance across diverse datasets (CIFAR100, ImageNet, MS-COCO, and TinyImageNe) and deep learning architectures (ResNet, VGG, ShuffleNet, MobileNet, WRN, and Faster-RCNN-FPN) and tasks (recognition, detection, and transfer learning).

## 2 RELATED WORKS

**Logit-based KD:** In the domain of KD, logit-based techniques have traditionally emphasized the distillation process utilizing solely the output logits. Historically, the primary focus of research

within logit distillation has been developing and refining regularization and optimization strategies rather than exploring novel methodologies. Noteworthy extensions to this conventional framework include the mutual-learning paradigm, frequently referenced as DML (Zhang et al., 2018), and incorporating the teacher assistant module, colloquially termed TAKD (Mirzadeh et al., 2020). Nonetheless, a considerable portion of the existing methodologies remain anchored to the foundational principles of the classical KD paradigm, seldom probing the intricate behaviors and subtleties associated with logits (Zhao et al., 2022). While the versatility of these logit-based methods facilitates their applicability across diverse scenarios, empirical observations suggest that their efficacy often falls short when juxtaposed against feature-level distillation techniques.

**Feature-based KD:** Feature distillation, a knowledge transfer strategy, focuses on utilizing intermediate features to relay knowledge from a teacher model to a student model. State-of-the-art methods have commonly employed this technique, with some working to minimize the divergence between features of the teacher and student models (Heo et al., 2019b;a; Romero et al., 2014). A richer knowledge transfer is facilitated by forcing the student to mimic the teacher at the feature level. Others have extended this approach by distilling input correlations, further enhancing the depth of knowledge transfer (Park et al., 2019; Tian et al., 2019; r21, 2019; Chen et al., 2021b). These methods, though high-performing, grapple with substantial computational demands and potential privacy issues, especially with complex models and large datasets. These challenges not only amplify processing time and costs but can also limit their practical applicability in real-world scenarios. Recognizing these challenges, our paper pivots its attention to logit-based distillation techniques.

**Applications with KD:** Rooted in foundational work by (Hinton et al., 2015) and further enriched by advanced strategies like Attention Transfer (Zagoruyko & Komodakis, 2016), ReviewKd (Chen et al., 2021b), Decoupled KD (Zhao et al., 2022) and other methods (Park et al., 2019; Tian et al., 2019), KD has significantly improved performance in core vision tasks, spanning recognition (Krizhevsky et al., 2012; Simonyan & Zisserman, 2014; He et al., 2016), segmentation(Qin et al., 2021; Liu et al., 2019), and detection (Li et al., 2022; Yang et al., 2022; Zheng et al., 2023; Xu et al., 2022). Beyond vision, KD has also made notable strides in NLP tasks like machine translation and sentiment analysis (Kim & Rush, 2016; Zhang et al., 2022). KD has proven valuable in addressing broader AI challenges, such as reducing model biases (Hossain et al., 2022; Chai et al., 2022; Zhou et al., 2021; Jung et al., 2021) and strengthening common-sense reasoning (West et al., 2021). We evaluate our method within the realms of image classification and object detection.

## 3 METHODOLOGY

### 3.1 KNOWLEDGE DISTILLATION REVISITED

Consider a set of distinct samples denoted as $\mathcal{X} = \{\mathbf{x}_i\}_{i=1}^n$, where $\mathbf{x}_i \in \mathbb{R}^m$ and $n$ represents the total number of samples. Given a parametric deep learning model $f_\theta$ with learnable parameters $\theta$, its output for a sample $\mathbf{x}_i$ is defined as $\mathbf{z}_i = f(\mathbf{x}_i)$, where $\mathbf{z}_i \in \mathbb{R}^c$, and $c$ denotes the number of classes within the sample set $\mathcal{X}$. In the context of Knowledge Distillation (KD) literature, the model's output $\mathbf{z}$ is often referred to as the logit of the model. For brevity, we will omit $\theta$ from the model notation $f$. To provide more context within the realm of knowledge distillation, we designate $f_T$ as the teacher model, and $f_S$ as the student model. The fundamental objective of KD is to minimize the divergence between the logits of the student and teacher for each sample in $\mathcal{X}$. This can be expressed mathematically as minimizing the objective, $L_{\text{KD}} = \sum_{\mathbf{x}_i \in \mathcal{X}} \ell\left(f_T(\mathbf{x}_i), f_S(\mathbf{x}_i)\right)$. Here, $\ell(\cdot, \cdot)$ is a loss function that measures the discrepancy between two vectors. For logit-based distillation, the primary objective is to align the softened logits of the student and teacher models. This alignment is quantified using the Kullback-Leibler (KL) divergence between the softened probabilities of the two models. Formally, the distillation loss, $L_{KD}$, is defined as:

$$L_{KD} = KL\left(\text{Softmax}\left(\frac{f_T(\mathbf{x}_i)}{\tau}\right) \,\middle|\middle|\, \text{Softmax}\left(\frac{f_S(\mathbf{x}_i)}{\tau}\right)\right) \tag{1}$$

Here, $\tau$ is the temperature parameter that modulates the softmax sharpness.

The primary hurdle with logit-based distillation lies in the fact that any logit vector $\mathbf{z}_i = f(\mathbf{x}_i)$ is considerably more compact than its feature vector counterpart. This makes it difficult to extract the wealth of information embedded in the teacher model (Romero et al., 2014). The following section outlines some potential limitations associated with logit-based knowledge distillation.

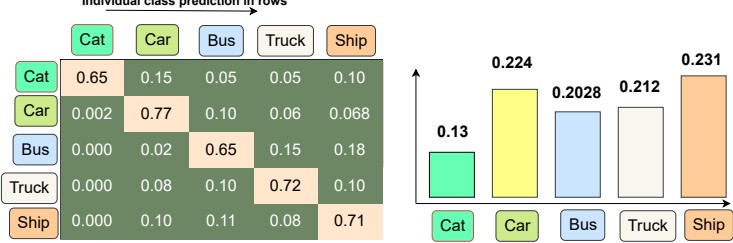

Figure 2: A toy example illustrating intra-class divergence. Each row of the prediction matrix (left) represents the mean prediction score of all images belonging to the corresponding class. By column-wise averaging, we can calculate an intra-class distribution (right) representing how different/similar a class is across other classes. Here, the 'cat' class differs from other vehicle classes (car, bus, truck, and ship). Thus, 'cat' gets a low intra-class score. We say this intra-class because this score averages the same class prediction across all images in the dataset. Previous KD methods do not transfer such intra-class variance from teacher to student during the distillation process.

**(1) Dilemma of single- vs. multi-instances:** While logit-based distillation primarily addresses single instances (Zhao et al., 2022; Zhang et al., 2018; Mirzadeh et al., 2020), these isolated logits lack insights regarding inter-relationships within a batch. To bridge this gap, Relational Knowledge Distillation (RKD) (Park et al., 2019) harnesses the power of inter-sample relationships by leveraging knowledge across data pairs $(\mathbf{x}_i, \mathbf{x}_j)$. However, in emphasizing such a relational perspective, RKD might omit the nuanced knowledge specific to individual samples. Later, to amplify instance-level knowledge transfer, Decoupled Knowledge Distillation (DKD) (Zhao et al., 2022) refines this approach by segregating logits into targeted (positive class) and non-targeted (negative class) components. Yet, while DKD improves precision, it remains isolated and does not establish interconnections among multiple images, potentially overlooking broader inter-sample dynamics.

**(2) Role of $\tau$:** In knowledge distillation, temperature scaling softens teacher model outputs, serving as a regularizer to reduce overfitting. Moreover, by preventing premature over-confidence in predictions, $\tau$ further promotes better generalization and reduces the risk of fitting too closely to training data (Hinton et al., 2015). Because of the teacher and student model's outputs have inherent statistical differences, finding a suitable value for $\tau$ is difficult (Liu et al., 2023). Usually, KD methods require extensive $\tau$ fine-tuning, leading to additional computational costs (Rafat et al., 2023).

**(3) Intra-class dependency:** We illustrate this issue in Figure 2. Irrespective of the input image, there is a distribution of the prediction scores of the teacher model for each individual class. This distribution reflects the teacher's positive bias toward the data. For example, as the class 'cat' is very different from other vehicle classes (car, bus, truck, and ship), a teacher can classify a 'cat' instance more confidently than other contemporary classes. No matter the input, the distribution of intra-class prediction (say 'cat') is ignored during the transfer of knowledge.

### 3.2 INTRODUCING *LumiNet*

Based on our previous discussion, the process of knowledge distillation is found to be further enriched for a given instance $\mathbf{x}_i$ when viewed in the context of its batch samples. Formally, a measure of information $K$ corresponding to $\mathbf{x}_i$ can be obtained as:

$$K(\mathbf{x}_i) \propto \mathcal{D}(\mathbf{x}_i) + \sum_{j \neq i} R(\mathbf{x}_i, \mathbf{x}_j) \qquad (2)$$

Here, $\mathcal{D}$ captures the inherent *dark knowledge*, while $R$ captures the inter-relational dynamics amongst instances. In this paper, we propose a loss function that optimizes this formulation.

**Constructing the perception:** We formulate our approach considering a batch of data samples $\mathcal{B} = \{\mathbf{x}_i\}_{i=1}^{b}$, which is randomly selected from the original dataset $\mathcal{X}$. Consequently, the logits generated by a model $f$ for an instance $\mathbf{x}_i \in \mathcal{B}$ across $c$ classes are represented as: $\mathbf{z}_i = (z_{i1}, z_{i2}, \dots, z_{ic})$, where $z_{ik}$ symbolizes the logit for the $j^{th}$ class for instance $x_i$. We adjust the logits based on the mean $U_j$ and variance $V_j$ of across each class $j$ of a batch. This transformed logit is given by: $h_{ij} = \frac{z_{ij} - U_j}{\sqrt{V_j}}$. Here, $h_{ij}$ represents the augmented logit for the $j^{th}$ class for instance $\mathbf{x}_i$.

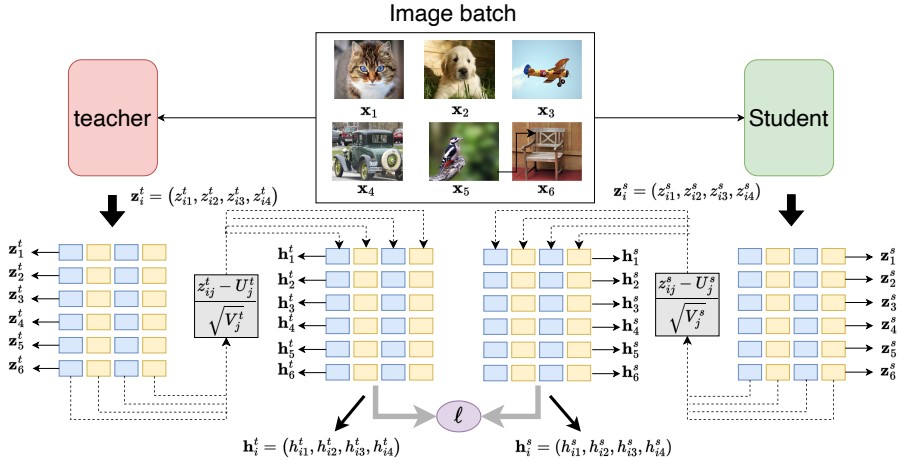

Figure 3: Given a batch of samples $\mathcal{B} = \{\mathbf{x}_1, \mathbf{x}_2, \mathbf{x}_3, \mathbf{x}_4, \mathbf{x}_5, \mathbf{x}_6\}$, both the teacher and student models generate logit for each sample in the batch, denoted as $\mathbf{z}_i^t = (z_{i1}^t, z_{i2}^t, z_{i3}^t, z_{i4}^t)$ and $\mathbf{z}_i^s = (z_{i1}^s, z_{i2}^s, z_{i3}^s, z_{i4}^s)$. In the subsequent stage, $mean((U_j^t, U_j^s))$ and $variance(V_j^t, V_j^s)$ for each class in the batch are computed for both teacher and student logit. These values are then used to normalize the logit of both models, resulting in a new logit representation referred to as the Perception logit: $\mathbf{h}_i^t = (h_{i1}^t, h_{i2}^t, h_{i3}^t, h_{i4}^t)$ and $\mathbf{h}_i^s = (h_{i1}^s, h_{i2}^s, h_{i3}^s, h_{i4}^s)$. Finally, a loss function $\ell$ is calculated between the teacher and student to complete the knowledge distillation process.

Consequently, the augmented logits for instance $\mathbf{x}_i$ are obtained as:

$$\mathbf{h}_i = \left( \frac{z_{i1} - U_1}{\sqrt{V_1}}, \frac{z_{i2} - U_2}{\sqrt{V_2}}, \dots, \frac{z_{ic} - U_c}{\sqrt{V_c}} \right) \tag{3}$$

In this context, the reconstructed logits $\mathbf{h}_i$ capture the model's perception. Instead of merely making raw predictions, both the models (teacher and student) try to understand the finer details and differences within the batch of data. As outlined in Eq. 3, the method of constructing 'perceived' logits is explained. In short, When both the teacher and Student models' intra-class predictions are adjusted on the same scale, keeping their variance constant, it influences the probability distribution across all the classes for individual instances. This new set of logits offers us a more insightful perspective of each instance. This set of logits $\mathbf{h}_i$ is referred to as 'perception'.

**The *LumiNet* Loss:** Classical knowledge distillation seeks to transfer the rich perceptual capabilities of a teacher model onto a smaller student model. To this end, we introduce *LumiNet*, a novel approach emphasizing the alignment of 'perceptions' rather than raw logits. In *LumiNet*, we focus on the perceived logits. Given an instance $\mathbf{x}_i$, we denote the logits from the teacher for class $c$ as $h_{ic}^t$ and those from the student as $h_{ic}^s$. The softmax operation scaled by a temperature factor $\tau$ produces probability distributions as: $P_c^T(\mathbf{x}_i) = \frac{\exp(h_{ic}^t/\tau)}{\sum_{c'} \exp(h_{ic'}^t/\tau)}$, $P_c^S(\mathbf{x}_i) = \frac{\exp(h_{ic}^s/\tau)}{\sum_{c'} \exp(h_{ic'}^s/\tau)}$. With this understanding, the *LumiNet* loss can be represented as:

$$\mathcal{L}_{LumiNet} = \sum_{\mathbf{x}_i \in \mathbf{X}} \sum_c P_c^T(\mathbf{x}_i) \log \frac{P_c^T(\mathbf{x}_i)}{P_c^S(\mathbf{x}_i)}, \tag{4}$$

The objective of the *LumiNet* loss Eq. 4 is to ensure that the student model not only aligns its predictions with the teacher but does so in the transformed 'perception' space. This ensures that the student does not just parrot the teacher's outputs but also learns a deeper understanding of intra-class and inter-class relationships. It also aligns with our intent outlined in Eq. 2. By minimizing the *LumiNet* loss, we ensure that the student model's perception of data instances closely mirrors that of the teacher's, leading to a more robust and nuanced student model.

### 3.3 THE BRIGHT SIDE OF *LumiNet* PERCEPTION

While conventional Knowledge Distillation frameworks face challenges in both logit-based and feature-based implementations, our methodology in *LumiNet* sheds light on a renewed perspective to tackle these issues. Here's a detailed exploration of the bright side of *LumiNet*'s approach to KD:

1. **Enhanced logit granularity with perception:** Traditional logit-based approaches are restricted by the inherent granularity of their representations, as characterized by the direct logits of any input $\mathbf{x}_i$. In contrast, *LumiNet*, leveraging its perception, refines this representation by introducing a transformation. Through the utilization of the mean $U_j$ and variance $V_j$ for the logits of each class within a batch, as defined in the perceived logits $\mathbf{x}_i'$ in Eq. 3, *LumiNet* achieves a more nuanced understanding. This mathematical recalibration allows the model to encapsulate subtler distinctions and depth, addressing the limitations inherent to conventional logit presentations.

2. **Balanced softening and overfitting:** In traditional knowledge distillation (KD), the temperature parameter $\tau$ tempers logits by pushing their values closer to zero, effectively reducing variance and bridging the gap between teacher and student logits for efficient knowledge transfer.In *LumiNet*, logits $\mathbf{x}_i'$ are intra-class normalized, yielding a zero mean and unit variance for each class. Thus, the reliance on $\tau$ for inter-class adjustments is diminished due to the intrinsically reduced variance and mean of the logits.

3. **Holistic KD approach:** As discussed in the previous section, although effective in its realm, The DKD (Zhao et al., 2022) methodology sometimes fails to wholly capture the essence of the teacher's knowledge. *LumiNet*, with its perception-driven paradigm, seamlessly amalgamates targeted and non-targeted knowledge, as shown in Eq. 4. This holistic approach ensures a broader and deeper transference of knowledge.

4. **Capturing inter-instance relationships:** Recognizing the essence of both intra and inter-class contexts, *LumiNet* employs transformations on logits through intra-class mean and variance computations to produce normalized logits $h_{ic}$. This process intrinsically captures intra-class dynamics. Concurrently, by considering logits across all classes for an instance $\mathbf{x}_i$ within the batch, *LumiNet* implicitly addresses inter-class relationships as well. Hence, with the formulations and variables defined in previous sections, *LumiNet* ensures that the nuances within a class and the broader inter-class relationships are effectively captured, enriching the learning context for the student model.

In essence, *LumiNet* redefines the horizons of logit-based KD. This innovative approach not only rectifies recognized challenges but also pioneers a roadmap on enhancing logit-based KD techniques to potentially overshadow their feature-based counterparts.

## 4 EXPERIMENTS

### 4.1 SETUP

**Dataset:** Using benchmark datasets, we conduct experiments on three vision tasks: image classification, object detection, and transfer learning. Our experiments leveraged *four* widely acknowledged benchmark datasets. First, **CIFAR-100** (Krizhevsky et al., 2009), encapsulating a compact yet comprehensive representation of images, comprises 60,000 32x32 resolution images, segregated into 100 classes with 600 images per class. **ImageNet** (Russakovsky et al., 2015), a more extensive dataset, provides a rigorous testing ground with its collection of over a million images distributed across 1,000 diverse classes, often utilized to probe models for robustness and generalization. Concurrently, the **MS COCO** dataset (Lin et al., 2014), renowned for its rich annotations, is pivotal for intricate tasks, facilitating both object detection and segmentation assessments with 330K images, 1.5 million object instances, and 80 object categories. We strictly adhered to the standard dataset splits for reproducibility and benchmarking compatibility for training, validation, and testing. The **TinyImageNet**[1] dataset, although more compact, acts as an invaluable resource for transfer learning experiments due to its wide variety across its 200 classes.

---

[1] https://www.kaggle.com/c/tiny-imagenet

Table 1: Recognition results on the CIFAR-100 validation.

| | (a) Same architecture | | | | | | (b) Heterogeneous architecture | | | | |
|---|---|---|---|---|---|---|---|---|---|---|---|
| **Teacher** | ResNet56 | ResNet110 | ResNet32×4 | WRN-40-2 | WRN-40-2 | VGG13 | ResNet32×4 | WRN-40-2 | VGG13 | ResNet50 | ResNet32×4 |
| | 72.34 | 74.31 | 79.42 | 75.61 | 75.61 | 74.64 | 79.42 | 75.61 | 74.64 | 79.34 | 79.42 |
| **Student** | ResNet20 | ResNet32 | ResNet8×4 | WRN-16-2 | WRN-40-1 | VGG8 | ShuffleNet-V1 | ShuffleNet-V1 | MobileNet-V2 | MobileNet-V2 | ShuffleNet-V2 |
| | 69.06 | 71.14 | 72.50 | 73.26 | 71.98 | 70.36 | 70.50 | 70.50 | 64.60 | 64.60 | 71.82 |
| **Feature-Based Methods** | | | | | | | **Feature-Based Methods** | | | | |
| FitNet | 69.21 | 71.06 | 73.50 | 73.58 | 72.24 | 71.02 | 73.59 | 73.73 | 64.14 | 63.16 | 73.54 |
| RKD | 69.61 | 71.82 | 71.90 | 73.35 | 72.22 | 71.48 | 72.28 | 72.21 | 64.52 | 64.43 | 73.21 |
| CRD | 71.16 | 73.48 | 75.51 | 75.48 | 74.14 | 73.94 | 75.11 | 76.05 | 69.73 | 69.11 | 75.65 |
| OFD | 70.98 | 73.23 | 74.95 | 75.24 | 74.33 | 73.95 | 75.98 | 75.85 | 69.48 | 69.04 | 76.82 |
| ReviewKD | 71.89 | 73.89 | 75.63 | 76.12 | 75.09 | 74.84 | 77.45 | 77.14 | 70.37 | 69.89 | 77.78 |
| FCFD | 71.68 | - | 76.80 | 76.34 | 75.43 | 74.86 | 78.12 | 77.81 | 70.67 | 71.07 | 78.20 |
| **Logit-Based Methods** | | | | | | | **Logit-Based Methods** | | | | |
| KD | 70.66 | 73.08 | 73.33 | 74.92 | 73.54 | 72.98 | 74.07 | 74.83 | 67.37 | 67.35 | 74.45 |
| DML | 69.52 | 72.03 | 72.12 | 73.58 | 72.68 | 71.79 | 72.89 | 72.76 | 65.63 | 65.71 | 73.45 |
| TAKD | 70.83 | 73.37 | 73.81 | 75.12 | 73.78 | 73.23 | 74.53 | 75.34 | 67.91 | 68.02 | 74.82 |
| DKD | 71.97 | 74.11 | 76.32 | 76.24 | 74.81 | 74.68 | 76.45 | 76.70 | 69.71 | 70.35 | 77.07 |
| **Ours** | **72.29** | **74.2** | **77.50** | **76.38** | **75.12** | **74.94** | **76.66** | **76.95** | **70.50** | **70.97** | **77.55** |
| Δ | +1.63 | +1.12 | +4.17 | +1.37 | +1.58 | +1.96 | +2.59 | +2.12 | +3.13 | +3.62 | +3.1 |

**Network architectures:** Various architectures are employed depending on the context. For CIFAR-100, homogeneous configurations use teacher models like ResNet56, ResNet110 (He et al., 2016), and WRN-40-2, paired with corresponding students such as ResNet20 and WRN-16-2 (Table 1a). In heterogeneous settings, architectures such as ResNet32×4 and VGG13 (Simonyan & Zisserman, 2014) for teachers are paired with lightweight models like ShuffleNet-V1, ShuffleNet-V2 (Ma et al., 2018) and MobileNet-V2 (Sandler et al., 2018) as students (Table 1b). For ImageNet classification, ResNet34 was employed as the teacher and ResNet18 as the student. Additionally, For object detection on MS-COCO, the Faster RCNN with FPN (Zhang et al., 2022) was utilized as the feature extractor, with predominant teacher models being ResNet variants, while the latter served as a student. A pre-trained WRN_16_2 model is further harnessed for transfer learning. We also conducted tests on the ViT model (Dosovitskiy et al., 2020). The DeiT-s(Touvron et al., 2021) was employed as the teacher model, while its more compact version was used as the student model.

**Evaluation metric:** We assess methods' performance using Top-1 and Top-5 accuracy for classification tasks. We employ Average Precision (AP, AP50, and AP70) to gauge precision levels in object detection tasks. We calculate a Δ that denotes the performance improvement of *LumiNet* over the classical KD method, underlining our approach's enhancements.

**Implementation details[2]:** We explore knowledge distillation using two configurations: a homogenous architecture, where both teacher and student models have identical architectural types (ResNet56 and ResNet20), and a heterogeneous architecture, where they differ (ResNet32x4 as the teacher and ShuffleNet-V1 as the student). Our study incorporates a range of neural network architectures such as ResNet, WRN, VGG, ShuffleNet-V1/V2, and MobileNetV2. Training parameters are set as follows: for CIFAR-100, a batch size of 64 and a learning rate of 0.05; for ImageNet, a batch size of 128 and a learning rate of 0.1; and for MS-COCO, a batch size of 8 with a learning rate of 0.01. The value of 't' varies from 1 to 8, depending on the specific model. We followed (Zhao et al., 2022). Because the logits are already smoothed out, it's good to use higher CE and KD settings than the classical settings. To implement distillation in the ViT variant, we adopted the implementation settings detailed by (Wang et al., 2022). All models are trained on a single GPU.

## 4.2 MAIN RESULTS

**Comparison methods:** We compare our method with well-established feature- and logit-based distillation methods, underscoring its potential and advantages in the knowledge distillation domain. Notable methods in *Feature Based Methods* category include FitNet (Romero et al., 2014), which aligns features at certain intermediary layers, RKD (Park et al., 2019) that focuses on preserving

---

[2]Codes and models are available at `TBA`

Table 2: ImageNet results. ResNet34 and ResNet18 serve as the teacher and student, respectively.

| | | | Feature-Based Methods | | | | Logit-Based Methods | | | | |
| --- | --- | --- | --- | --- | --- | --- | --- | --- | --- | --- | --- |
| | Teacher | Student | AT | OFD | CRD | ReviewKD | KD | DML | TAKD | DKD | **Ours** |
| Top-1 | 73.31 | 69.75 | 70.69 | 70.81 | 71.17 | 71.61 | 70.66 | 70.82 | 70.78 | 71.70 | **71.78** |
| Top-5 | 91.42 | 89.07 | 90.01 | 89.98 | 90.13 | 90.51 | 89.88 | 90.02 | 90.16 | 90.41 | **90.71** |

Table 3: Comparison of training time per batch, number of extra parameters ($\theta$) on the CIFAR-100.

| Teacher: ResNet32×4
Student: ResNet8×4 | KD | RKD | FitNet | OFD | CRD | ReviewKd | DkD | **Ours** |
| --- | --- | --- | --- | --- | --- | --- | --- | --- |
| Latency↓ (ms) | 11 | 25 | 14 | 19 | 41 | 26 | 11 | **11** |
| $\theta$ ↓ | 0 | 0 | 16.8k | 86.9k | 12.3M | 1.8M | 0 | **0** |
| Acc↑ (%) | 73.33 | 71.90 | 73.50 | 74.95 | 75.51 | 75.63 | 76.32 | **77.50** |

Table 4: Detection results on MS-COCO using Faster-RCNN-FPN (Lin et al., 2017) backbone.

| T: ResNet110, S: ResNet18 | | | Feature-Based Methods | | | Logit-Based Methods | | | |
| --- | --- | --- | --- | --- | --- | --- | --- | --- | --- |
| | Teacher | Student | FitNet | FGFI | ReviewKD | KD | TAKD | DKD | **Ours** |
| AP | 42.04 | 33.26 | 34.13 | 35.44 | **36.75** | 33.97 | 34.59 | 35.05 | 35.34 |
| $AP_{50}$ | 62.48 | 53.61 | 54.16 | 55.51 | 56.72 | 54.66 | 55.35 | 56.60 | **56.82** |
| $AP_{75}$ | 45.88 | 35.26 | 36.71 | **38.17** | 34.00 | 36.62 | 37.12 | 37.54 | 37.56 |

pairwise relations of examples, and CRD (Tian et al., 2019) which minimizes the contrastive loss between the representations of the teacher and student models. Other methods in this category include OFD (Cho & Hariharan, 2019) and ReviewKD (Chen et al., 2021b), each bringing unique strategies to leverage intermediary network features. *Logit Based Methods* methods include KD (Hinton et al., 2015), DML (Zhang et al., 2018), TAKD (Mirzadeh et al., 2020), and DKD (Zhao et al., 2022), which ensures that the student's logits are similar to the teacher.

**Recognition tasks:** We perform image recognition tasks on CIFAR-100 and ImageNet. On **CIFAR-100**, when teacher and student models shared identical architectures, shown in Table 1a, *LumiNet* presented improvements of 2-3%. And when the architectures were from different series, shown in Table 1b, the improvements were between 3-4%, consistently outperforming the baseline, classical KD, and other methods rooted in KD's principles. Similarly, on the intricate **ImageNet** dataset, *LumiNet* outshined, exceeding all logit-based distillation techniques and besting state-of-the-art feature-based distillation methods, shown in Table 2. These findings robustly demonstrate that regardless of dataset or architectural variations, *LumiNet* maintains its unparalleled efficacy and highlights its unique capacity to learn based on the 'perception'. In Table 3, we show that *LumiNet* showcases a superior trade-off between the number of extra parameters/running time vs. accuracy. The necessity for extra parameters in feature-based techniques arises from integrating projection or intermediate layers, which align the teacher's feature space to the student model. With a latency of 11 ms, our method matched the best-performing models in speed and is exceptionally efficient (77.50%) without extra parameters. This combination of low latency, less computation, and high accuracy further reaffirms the unparalleled effectiveness and efficiency of the *LumiNet*.

**Detection task:** The quality of deep features is pivotal for accurate object detection. One persistent challenge is the effective knowledge transfer between established teacher models and student detectors (Li et al., 2017). Generally, logits cannot provide knowledge for object localization (Wang et al., 2019). While logit-based techniques have been traditionally used for this, they often do not meet state-of-the-art standards. On **MS COCO** dataset, *LumiNet* delivered noticeably better results (Table 4) compared to logit-based methods, which is comparable to feature-based methods.

**Transfer learning task:** To assess the transferability of deep features, we conduct experiments to verify our algorithm's superior generalization capabilities *LumiNet*. In this context, we utilized the Wide Residual Network (WRN-16-2), distilled from WRN-40-2, as our principal feature extraction apparatus. Sequential linear probing tasks were subsequently performed on the benchmark downstream dataset, notably *Tiny-ImageNet*. Our empirical results, delineated in Figure 4(a), manifestly underscore the exemplary transferability of features cultivated through *LumiNet*.

**Effect of Strong Augmentation:** In Table 5, we report the performance after using auto augmentation by increasing the complexity of training samples (Cubuk et al., 2019). *Luminet* outperforms auto augmentation-based method (Jin et al., 2023) in heterogeneous and homogeneous on the CIFAR-100 dataset. The results show our effectiveness in distilling knowledge from challenging samples.

**Vision Transformer:** To explore the capabilities of Luminet beyond conventional Convnet models, we conducted knowledge distillation from a DeiT-s (ViT variants) to its compact variant. The results in Table 6 indicate that Luminet surpasses traditional KD approaches. This demonstrates Luminet's ability to extract knowledge across varied settings effectively.

Table 5: Results after applying auto Augmentation.

| Teacher | WRN_40_2 | WRN_40_2 | VGG 13 | ResNet32×4 | WRN-40-2 |
|---|---|---|---|---|---|
| Accuracy | 75.61 | 75.61 | 74.64 | 79.42 | 75.61 |
| Student | WRN_16_2 | WRN_40_1 | VGG 8 | ShuffleNet-V2 | ShuffleNet-V1 |
| Accuracy | 73.26 | 71.98 | 70.36 | 71.82 | 70.50 |
| Jin et al. (2023) | 76.63 | 75.35 | 75.18 | 78.44 | 77.44 |
| Ours | **76.91** | **76.01** | **75.57** | **79.12** | **77.97** |
| Δ | +0.28 | +0.66 | +0.39 | +0.68 | +0.53 |

Table 6: Performance using ViT

| Algorithm | params | CIFAR-100 |
|---|---|---|
| Teacher | 21.3M | 76.30% |
| Student | 2.38M | 65.46% |
| KD | 2.38M | 67.38% |
| Ours | 2.38M | 68.42% |

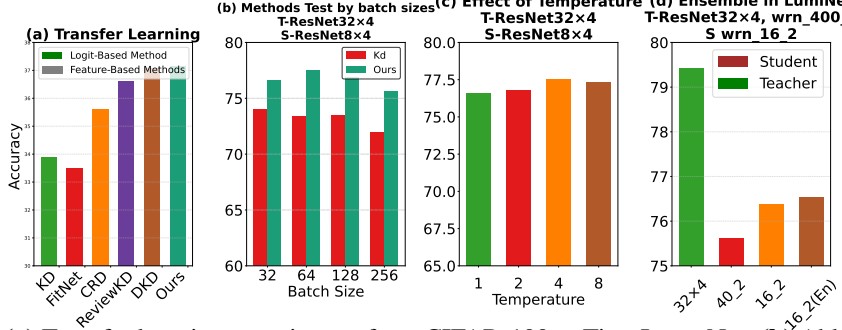

Figure 4: **(a)** Transfer learning experiments from CIFAR-100 to Tiny-ImageNet. **(b)** Ablation study on different batch sizes. **(c)** Impact of different $\tau$ values. **(d)** Performance on ensemble learning.

## 4.3 ABLATION STUDY

**Varying batch sizes:** Figure 4(b) showcases an ablation study comparing the performance of the *LumiNet* method with both a basic student model and the Knowledge Distillation method across various batch sizes. The batch sizes range from 32 to 256. The student model, serving as a standard baseline, demonstrates a slight decline in performance as the batch size increases. In comparison, *LumiNet* consistently outperforms both the student and Kd methods across all tested batch sizes, suggesting its robustness and superiority in the given context.

**Varying $\tau$:** The logits within our perception framework are reconstructed with a clear statistical understanding intra-class logits. For this, both the teacher and the student models exhibit "softened" values, achieved through normalization by variance and maintaining an intra-class mean of zero. Consequently, the dependency on temperature $\tau$ is minimal. Empirical evaluations in Figure 4(c) suggest minimal performance fluctuations across $\tau$ (ranging between 1 and 8) yield optimal results.

**Ensemble of teachers:** We employ an ensemble of two teacher models: ResNet 8x4 and WRN-40-2 (labeled in the figure as "8x4" and "40-2"). This ensemble technique, which we term "Logit Averaging Ensemble," involves averaging the logits produced by the two teacher models (Sagi & Rokach, 2018). When training the student model, WRN-16-2 (labeled as "16-2" for the regular student and "16-2(en)" for the student learned by ensemble technique), we observed a notable improvement in accuracy using this ensemble-derived guidance. As shown in Figure 4(d), when conventionally train with our *LumiNet* approach with just the WRN-40-2 teacher, we achieve 76.38% accuracy. However, results improve slightly to 76.52% when the training is augmented with insights from the ensemble technique. This suggests that the ensemble's aggregated information potentially enables the student model to capture more intricate patterns and nuances from the teachers.

## 5 CONCLUSION

We propose a novel logit-based knowledge distillation strategy. Our study in KD underscores the pivotal role of intra-class variations—a dimension often underemphasized in prevailing methods. Within these class structures lie nuanced insights that traditional methods (overly reliant on replicating teacher logits) might overlook. We present *LumiNet*, a strategy imbued with the 'perception' concept, as a new solution to address this challenge. Supported by empirical results from our rigorous experiments across recognition, detection, and transfer learning, *LumiNet* consistently demonstrates superior efficacy. In essence, for knowledge distillation to truly shine and be comprehensive, it is essential to illuminate and address these intra-class disparities. Our research with *LumiNet* brings forth the bright side of perceptual knowledge distillation, guiding the way on this path.

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
