# SUPPLEMENTARY MATERIAL FOR "LUMINET: THE BRIGHT SIDE OF PERCEPTUAL KNOWLEDGE DISTILLATION"

## ABSTRACT

This supplementary material provides additional details in support of the contribution presented in the main paper.

- Section 1: Alternative Perspective on Instance-Based Knowledge Distillation Method. (additional discussion in support of Section 3.3 of the main paper)

- Section 2: Incorporating with feature-based distillation.(additional discussion in support of Section 4.2 of the main paper)

# 1 ALTERNATIVE PERSPECTIVE ON INSTANCE-BASED KNOWLEDGE DISTILLATION METHOD.

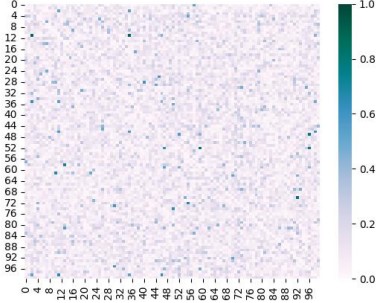

Figure 1: Comparing Teacher-Student Predictions Similarity in the DKD Method

Figure 2: Comparing Teacher-Student Predictions Similarity in Our Method

The primary goal of instance-based knowledge distillation is to replicate the raw logits of the teacher, as illustrated in Figure 1. This figure demonstrates that the predictions closely resemble the teacher's logits. However, in various contexts, such as heterogeneous architectures or object detection techniques, instance-based methods struggle to effectively capture optimal knowledge. In our approach, *LumiNet*, depicted in Figure **??**, the prediction similarity to the teacher's logits is significantly lower compared to DKD. Yet, as detailed in the main paper, *LumiNet* achieves better performance scores than DKD(Zhao et al., 2022).

This indicates that in instance-based methods, we can diverge from the traditional approach of closely imitating the raw logits of teacher models. Instead, we can focus on extracting relational knowledge by applying a perception matrix within instance-based settings. This approach enables the student model to learn independently without directly mimicking the features or raw logits of the teacher model.

Table 1: Performance of our method with the incorporation of ReviewKD Loss

| Teacher/Student Architecture | ReviewKD | Ours | Ours* |
|---|---|---|---|
| WRN-40-2 → ShuffleNet-V1 | 77.14 | 76.95 | **77.29** |
| ResNet32×4 → ShuffleNet-V2 | 77.78 | 77.55 | **77.93** |

## 2    INCORPORATING WITH FEATURE-BASED DISTILLATION

In our experiments, we typically do not use feature-based distillation loss. However, to enhance performance in certain architectures, we combined feature-based loss from ReviewKD (Chen et al., 2021) with our *LumiNet* loss. This integration resulted in improved performance, as demonstrated in Table 1. However, we note that feature-based methods raise concerns such as privacy risks in adversarial attacks and increased computational demands. Despite these challenges, our experiment demonstrates that combining feature-based losses enhances our method's performance.

We also examined constraints in feature-based distillation methods, finding that they often lead to longer convergence times. For instance, ReviewKD (Chen et al., 2021), despite its comprehensive approach, requires significant training time due to its multi-level distillation process and complex components like the Attention-Based Fusion module. OFD (Cho & Hariharan, 2019), while focusing on multi-layer distillation, demands extra convolutions for feature alignment, increasing computational needs. Similarly, CRD (Tian et al., 2019) employs a contrastive loss that requires a large memory bank, adding to computational costs.

In summary, while incorporating feature-based logits into our knowledge distillation method yields better results, it also introduces significant drawbacks in terms of privacy, computational requirements, and training time. Hence, we advocate for logit-based knowledge distillation as a more resource-efficient and versatile alternative for various applications.