# OpenReview forum: "LumiNet: The Bright Side of Perceptual Knowledge Distillation"
_ICLR.cc/2024/Conference — Submitted to ICLR 2024_

### Official Review · Reviewer_8pKb · 2023-10-23

**Soundness:** 2 fair
**Presentation:** 3 good
**Contribution:** 2 fair
**Rating:** 5
**Confidence:** 3

**Summary:**

This paper proposes a logit-based KD method through the reconstruction of instance-level distributions. The proposed LumiNet emphasizes the perception alignment instead of the traditional logit output. Through this framework, the student model can learns the inter-class and intra-class relationship from the teacher model. Extensive experiment results prove that the proposed method shows competitive performance in both image classification and detection.

**Strengths:**

1. This paper is well-written and easy to follow.
2. This paper conducts extensive experiments in both image recognition and detection.

**Weaknesses:**

1. Capturing inter-instance and inter-class relationship to boost knowledge distillation is not new to me. For example, the idea of [1] is quite similar with this paper. I want to see more explanations during rebuttal period.
[1] Multi-Level Logit Distillation (CVPR2023)
2.  Missed baseline TAKD and DML in experiment section.

**Questions:**

See weakness. I think the authors should have more explanations on the novelty of their method.

---

> ### Author Response · Authors · 2023-11-23
>
> Thank you for the positive feedback on our paper. We're glad that our experiments were well-received and found to be of value.
>
> # Inter-instance and inter-class relationship
>
> Here, we explain how our method differs from other contemporary approaches and its unique effectiveness.
>
> ## Inter-class relationship
>
> Traditional logit-based knowledge distillation (KD) methods [1,2,3,4], primarily adopt an instance-based approach. DKD leverages this approach effectively by incorporating additional losses alongside the standard cross-entropy loss. This method is adept at capturing inter-class relationships for individual instances, thus effectively utilizing 'dark knowledge' to enhance performance. While instance-based knowledge typically focuses on the output of the last layer (e.g., soft targets), it often overlooks the vital intermediate-level supervision provided by the teacher model. This supervision is crucial for representation learning, especially in very deep neural networks.
>
> ## Inter-instance relationship
>
> Inter-instance relation-based knowledge delves into the relationships between different layers or data samples. To find a better relation between two layers,  Yim et al. (2017)[5] introduced FSP, which uses the Gram matrix between two layers to summarize the relations between pairs of feature maps, calculated using inner products between features from these layers. Most of the methods extract knowledge from **hint (feature-based) layers** [6], which is the output of a teacher's hidden layer. For this, these methods become feature-based and expose privacy concerns for adversarial attacks, computational requirements, etc. We notice that most of the methods do not consider the inter-class and instance-based relations together. In this paper, we aim to integrate both inter-class and inter-instance relationships but without the help of feature layers so that our method becomes logit-based KD.
>
> ## LumiNet vs. Jin et al. "Multi-Level Logit Distillation"
>
> Our method and Jin et al. share similar motivations, i.e., considering both inter-class and inter-instance relationships from output predictions (logit layers). To extract knowledge, Jin et al. introduced batch-, class- and instance-level alignment. For batch- and class-level alignments, they utilized the well-known Gram matrix and correlation methods, and importantly, they introduced **three additional loss functions**. Like Jin et al., our method emphasizes output prediction to understand subtle knowledge. However, our method simplifies the process by not requiring additional loss functions to understand inter-class relationships. We have introduced a new concept called the "perception" matrix (Eq. 3), distinct in its mathematical approach from previous methods, aimed at capturing inter-instance relationships. This matrix is tailored to enhance and adjust the logits on an instance level.
>
> For improved representation in Jin et al., prediction augmentation is utilized, which involves augmenting the instance-level representation by dividing it with a temperature value. However, our approach does not require such a strategy. Notably, this algorithm heavily depends on the auto (strong) augmentation technique, which is discussed in detail in response to reviewer 3qnl.
>
> The fundamental differences in approach and philosophy to knowledge distillation between LumiNet and Multi-Level Logit distillation are significant. While LumiNet focuses on recalibrating instance-level logits using a perception matrix of inter-class instances, Multi-Level Logits emphasizes direct alignment and linear augmentation across multiple levels by introducing additional losses and complexities.
>
> # Including baseline TAKD and DML
>
> Thanks for pointing this out. TAKD and DML results are already in Tables 1 and 4. In the revision, we have included those results in Tables 2 and 3. Now, all baselines are considered in the experimental section.
>
> **References**
> 1. Hinton, Geoffrey, Oriol Vinyals, and Jeff Dean. "Distilling the knowledge in a neural network." arXiv 2015.
> 2. Zhang, Ying, et al. "Deep mutual learning." CVPR. 2018.
> 3. Mirzadeh, Seyed Iman, et al. "Improved knowledge distillation via teacher assistant." AAAI 2020.
> 4. Zhao, Borui, et al. "Decoupled knowledge distillation." cvpr. 2022.
> 5. Yim, Junho, et al. "A gift from knowledge distillation: Fast optimization, network minimization and transfer learning." CVPR 2017.
> 6. Romero, Adriana, et al. "Fitnets: Hints for thin deep nets." arXiv 2014.

---

### Official Review · Reviewer_tkRa · 2023-10-30

**Soundness:** 2 fair
**Presentation:** 2 fair
**Contribution:** 2 fair
**Rating:** 5
**Confidence:** 5

**Summary:**

In this work, the authors use statistic matrixes to recalibrate logits for knowledge distillation, named LumiNet.  LumiNet focuses on inter-class relationships and enables the student model to learn a richer breadth of knowledge. Both teacher and student models are mapped onto the statistic matrixes, with the student’s goal being to minimize representational discrepancies.

**Strengths:**

1. Distillation is an important topic to our community, the proposed method is simple.
2. The write-up is easy to understand.

**Weaknesses:**

1. My major concern is the generalization, given that distillation is a well-defined topic, but this method seems like doesn't work well in heterogeneous architecture settings. I know it's a logit-based method, but the authors claimed to bridge this gap. Maybe the authors can combine feature-based loss in their method to see if their proposed method does work or not.

2. More state-of-the-art methods should be compared in this work. e.g. [1]


[1] Liu, Dongyang, Meina Kan, Shiguang Shan, and Xilin Chen. "Function-consistent feature distillation." (ICLR 2023)

**Questions:**

If this method get any benefit when combined with feature-based loss?

---

> ### Author Response · Authors · 2023-11-23
>
> Thank you for your valuable feedback on our work. We are delighted to hear that you found our proposed method for distillation both important and straightforward.
>
> # Performance on heterogeneous architectures
>
> ## Table D. Comparison of ReviewKD, DKD, and Our Method in Heterogeneous Architectures, highlighting the performance differences
>
> | Teacher/Student Architecture | ReviewKD | DKD   | Ours  | Δ (DKD - ReviewKD) | Δ (Ours - ReviewKD) |
> |:----------------------------:|:--------:|:-----:|:-----:|:-------------------:|:--------------------:|
> | ResNet32×4 → ShuffleNet-V1   |   77.45  | 76.45 | 76.66 |        -1.00        |        -0.79         |
> | WRN-40-2 → ShuffleNet-V1     |   77.14  | 76.70 | 76.95 |        -0.44        |        -0.19         |
> | VGG13 → MobileNet-V2         |   70.37  | 69.71 | 70.50 |        -0.66        |        +0.13         |
> | ResNet50 → MobileNet-V2      |   69.89  | 70.35 | 70.97 |        +0.46        |        +1.08         |
> | ResNet32×4 → ShuffleNet-V2   |   77.78  | 77.07 | 77.55 |        -0.71        |        -0.23         |
>
> Here, we compare our method with leading state-of-the-art logit-based distillation (DKD) and feature-based method (ReviewKD) on heterogeneous architecture settings. Checking Δ (DKD - ReviewKD) one can see DKD outperforms ReviewKD in a **single** case (ResNet50 → MobileNet-V2). In contrast, observing Δ (Ours - ReviewKD) we see that our method beats ReviewKD in **two** scenarios (ResNet50 → MobileNet-V2 and VGG13 → MobileNet-V2). On the other hand, when ReviewKD outperforms both DKD and our method, DKD gets performance gaps **1.00, 0.44, and 0.71**. In contrast, corresponding gaps are minimized in our case with **0.79, 0.19, and 0.13**. All this data shows that our method bridges the gap between logit- and feature-based methods more successfully than the previously known best logit-based distillation, DKD. In other words, While DKD tends to lag behind feature-based methods in various architectures, our approach has successfully reduced this performance disparity. From this, we can conclude our method achieves better generalization than DKD.
>
>
> # Incorporating Feature-Based Loss
>
> ##Table E. Performance of our method with incorporating to ReviewKD Loss
> | Teacher/Student Architecture | ReviewKD | Ours   | Ours*  |
> |------------------------------|----------|--------|--------|
> | WRN-40-2 → ShuffleNet-V1     | 77.14    | 76.95  | 77.29  |
> | ResNet32×4 → ShuffleNet-V2   | 77.78    | 77.55  | 77.93  |
>
> Here, we combine feature-based loss from ReviewKD in our method. As expected, this integration improved the performance of knowledge distillation, as shown in the table. We know that feature-based methods may impose privacy concerns for adversarial attacks, computational requirements, and so on. Therefore, we aim to advocate for logit-based knowledge distillation to make the process resource-efficient and versatile for various applications. However, this experiment shows that **one can improve the performance of our method by combining feature-based losses** and also considering those negative aspects of feature-based distillation. These results are added in the supplementary material.
>
> With the addition of other constraints in feature-based distillation methods, here, we point out loss function-specific details that may contribute to longer converge time.
>
> - **ReviewKD**: Employs a comprehensive knowledge distillation approach but demands significant training time due to its intricate multi-level distillation process and the integration of complex components like the Attention-Based Fusion module. In addition, the iterative review and fusion processes inherent in the technique extend the training duration.
>
> - **OFD**: Focuses on distilling information across multiple intermediate layers, yet it necessitates additional convolutions for feature alignment, leading to increased computational demands.
>
> - **CRD**: Introduces a contrastive loss to transfer pairwise relationships effectively, but this method requires maintaining a large memory bank for features (e.g., all 128-d features of ImageNet images), significantly elevating the computational costs.
>
> # Comparing with more state-of-the-art results
>
> In light of your feedback, we have added Liu et al. "Function-consistent feature distillation" for comparison with our method in Table 1.

---

### Official Review · Reviewer_7mGH · 2023-11-03

**Soundness:** 2 fair
**Presentation:** 2 fair
**Contribution:** 2 fair
**Rating:** 5
**Confidence:** 3

**Summary:**

This paper presents a new method for logit-based knowledge distillation. The authors define a perception matrix to recalibrate logits and use the recalibrated logits to perform knowledge.

**Strengths:**

- The method looks more effective than existing logit-based knowledge distillation techniques on various vision tasks.  The results of more complex object detection tasks are also provided.

- The method can improve performance without introducing extra parameters and latency.

**Weaknesses:**

- The improvements over DKD overall are not very significant. The proposed method and DKD share the general idea of modeling inter-instance relations to improve knowledge distillation. The method's performance improvements over DKD are limited in several cases, and it is as efficient as DKD. While it is good to see a solid improvement over DKD, the similarity between the two methods indeed limits the paper's contribution.

- The method is not evaluated on larger/stronger models. Will the method work well on popular vision Transformer architectures? Will the improvement be less significant if stronger teacher models are considered?

- I am not totally convinced that holistic methods like DKD "sometimes fail to wholly capture the essence of the teacher’s knowledge" but the proposed method can resolve this issue. What is "the essence of the teacher’s knowledge" here? In the paper, I can only find some intuitive analysis and overall performance to support the proposed method. Why the transformation in Eq.3 can solve this issue? Can you provide more specific analyses on why and when the existing methods fail and why the proposed method is better?

**Questions:**

Overall, I think it is a solid contribution to logit-based knowledge distillation. However, the core idea of the proposed method is not completely new and the improvement is not that significant. The paper also didn't provide an in-depth analysis of why the method is better.   I think it is an okay paper but not a strong/impressive contribution to the community. Therefore, I would like to rate this paper as "marginally below the acceptance threshold".

A minor issue: "... based on context and past encounters Johnson (2021)" -> "... based on context and past encounters (Johnson, 2021)"

----- Post rebuttal -----

I would like to first thank the authors for the detailed feedback. My concerns about the improvements over DKD have been properly addressed. However, I am still confused about the concept of "perception" introduced in the method. I think the new method is basically close to DKD and "Multi-Level Logit Distillation" mentioned by other reviewers. I am also not convinced by the explanation of the "unique" part of motivation and the advantages of the proposed method. It seems the authors want to create a new concept while failing to clearly illustrate why the new concept is necessary and there is also no empirical evidence to directly support the new concept.

After reading the authors' feedback as well as other reviews, I think there are still unresolved issues about the novelty, presentation, and experimental analysis. Therefore, I would like to keep my initial rating.

---

> ### Author Response · Authors · 2023-11-23
> **Addressing Weakness 1**
>
> Thank you for your insightful comments regarding the effectiveness of the method in comparison to existing logit-based knowledge distillation. Your observation is greatly appreciated.
>
> # Improvement over DKD
>
> In light of your comments, we have conducted a more detailed analysis of LumiNet's performance relative to DKD. Our revised table below presents this comparison across various architectures:
>
> ## Table C. Comparison of ReviewKD, DKD, and Our Method in similar Architectures, highlighting the performance differences
>
> |
> | **Teacher/Student**               | DKD    | ReviewKD | Ours   | Δ (DKD - ReviewKD) | Δ (Ours - ReviewKD) | Δ (Ours - DKD) |
> |-----------------------------------|--------|----------|--------|-------------------|---------------------|----------------|
> | **ResNet56 / ResNet20**           | 71.97  | 71.89    | 72.29  | +0.08             | +0.40               | +0.32          |
> | **ResNet110 / ResNet32**          | 74.11  | 73.89    | 74.20  | +0.22             | +0.31               | +0.09          |
> | **ResNet32×4 / ResNet8×4**        | 76.32  | 75.63    | 77.50  | +0.69             | +1.87               | +1.18          |
> | **WRN-40-2 / WRN-16-2**           | 76.24  | 76.12    | 76.38  | +0.12             | +0.26               | +0.14          |
> | **WRN-40-2 / WRN-40-1**           | 74.81  | 75.09    | 75.12  | -0.28             | +0.03               | +0.03          |
> | **VGG13 / VGG8**                  | 74.68  | 74.84    | 74.94  | -0.16             | +0.10               | +0.10          |
>
> *Note: Δ (DKD - ReviewKD), Δ (Ours - ReviewKD), and Δ (Ours - DKD) represent the performance differences between the methods.*
>
> As seen in Tables C and D (responded to reviewer tkRA ), LumiNet demonstrates consistent improvement over DKD across every architecture. It's noteworthy that while DKD does show improvement over KD, LumiNet further enhances these results. Additionally, when comparing DKD with ReviewKD, which often ranks second-best, DKD's advantage is marginal in some architectures and even less in others. This is particularly noteworthy, considering several logit-based KD methods tend to underperform relative to KD.
>
> ## DKD vs. Luminet
>
> In the traditional KD method, we generate a set of predictions (called a logit vector) for each training sample from both the teacher and student models. The goal is to make the students' predictions closely match the teacher's. This is done using KL divergence loss generally, which measures how one probability distribution diverges from a second, expected probability distribution. The methodology of DKD emerged from the traditional KD. The main objective of DKD is to match the student's logits as closely as possible with the teachers (shown in Fig. 4 of the DKD paper). The logit vector is divided into two parts: target (the term "TCKD" as mentioned in the DKD paper) and non-target (the term "NCKD" as mentioned in the DKD paper). The target part relates to the specific category we're interested in, while the non-target part contains other categories. DKD treats these two parts differently, using separate objective functions for each. The non-target categories hold 'dark knowledge'—subtle information crucial for the student model to capture inter-class relationships. By separating the logits, DKD aims to extract more detailed and valuable knowledge, potentially leading to better performance of the student model. In the final loss (Eq. 6 of DKD paper),  DKD sums up these two object functions and cross-entropy loss.
>
> Our approach differs fundamentally from DKD and other logit-based methods in a crucial aspect: **Instead of mimicking the raw logits from the teacher model, we've developed an innovative method named 'perception' to reconstruct the logits.** This reconstructed form is distinct from the raw logits. The reconstructed logits represent a different form, distinct from their original configuration. This reconstruction depends on the context of other instances in the same batch, differing significantly from traditional methods. Second, we emphasize the importance of intra-class relationships in recalibrating logits, a consideration absent in DKD and any instance-based KD algorithm. Lastly, unlike DKD's dual objective functions for separated logit vectors, we employ a single objective function combined with cross-entropy loss, challenging the need for logit decoupling that DKD claimed.

---

> ### Author Response · Authors · 2023-11-23
> **Addressing Weakness 2,3**
>
> # Effectiveness of our method on ViT and larger model as a teacher
>
> We assume that referring to stronger/larger teacher models means larger teacher models than the student model, as there is no universally defined strong teacher model. In this case, we have tested our method with significantly larger teacher models (table 1) compared to the student model. For instance, we have demonstrated results using combinations like (ResNet110, ResNet32) (vgg13, MobileNet V2) and (ResNet32x4, ShuffleNetV2), where the teacher models are considerably larger than the student models. In these scenarios, our method continued to perform at a state-of-the-art level, underscoring its effectiveness even with substantial differences in model size.
>
> Previous methods usually do not report results on vision transformer-based teacher models. However, we have conducted preliminary experiments in this area to address the reviewer's comment. We compared our results with traditional Knowledge distillation in Table B of the reviewer response 3qnL, which indicates that our method still maintains a similar level of performance advantage as observed in traditional KD setups.
>
> # Specific analyses Ours vs. existing methods
>
> In traditional knowledge distillation (KD), the primary aim is to comprehend the inter-class relationships of a specific instance. However, in the context of Differentiable Knowledge Distillation (DKD), although the objectives align closely with KD (as evidenced by the prediction similarity in Fig. 4 of the DKD paper), challenges arise in heterogeneous settings and complex tasks like object detection. That's why we mention 'sometimes fails to wholly capture the essence of teacher's knowledge'.  Instance-based knowledge traditionally concentrates on the outputs from the final layer, such as soft targets, but tends to neglect the crucial guidance offered by the teacher model at intermediate stages. This intermediate-level supervision is particularly important for learning representations in very deep neural networks [2].
>
> To address this, techniques based on features or relations have been employed [2, 3]. However, these approaches often require substantial resources and encounter various issues, as elaborated in response to Reviewer 8pKb. Our method introduces an innovative solution to this problem. We have developed a new function, which we term 'perception', derived from Eq. 3. This function enables the capture of relational knowledge from a batch without additional objective functions or resources.
>
> The key distinction between DKD and our method lies in the reconstruction of logits. In our approach, logits are restructured to carry information about other instances inherently. This novel advancement in instance-based distillation allows for more efficient and effective knowledge transfer without the need for extra computational resources or unnecessary complexity. In short, by employing Eq. 3, we have successfully integrated relational knowledge into the instance-based method, filling a previously existing gap.
>
> # Minor issues
>
> We have addressed other minor issues in the revised manuscript.
>
> **References**
> 1. Wang, Jiahao, et al. "Attention probe: Vision transformer distillation in the wild." ICASSP 2022
> 2. Yim, Junho, et al. "A gift from knowledge distillation: Fast optimization, network minimization and transfer learning." CVPR 2017.
> 3. Romero, Adriana, et al. "Fitnets: Hints for thin deep nets." arXiv 2014.

---

### Official Review · Reviewer_3qnL · 2023-11-04

**Soundness:** 2 fair
**Presentation:** 2 fair
**Contribution:** 2 fair
**Rating:** 5
**Confidence:** 5

**Summary:**

This paper proposes a new logit-based distillation method to distill a teacher model with only predicted logits. The proposed method, LumiNet,  can reconstruct more granular inter-class relationships, enabling the student model to learn a richer breadth of knowledge. The proposed LumiNet is evaluated on CIFAR-100, ImageNet, and MSCOCO, revealing its competitive performance.

**Strengths:**

1. The paper is clear, and the method is easy to follow.
2. LumiNet makes a simple but effective modification to logit KD, which calibrates the mean and variance of each class.

**Weaknesses:**

1. A strong baseline [1] should be discussed and compared. It seems that [1] also tries to learn more granular relationships, which achieves better performance than LumiNet in many benchmark results. If LumiNet can directly outperform [1] or further improve performance based on [1] should be discussed.
2. The experiments are performed on Conv-based methods. It will be convincing that LumiNet can show its advantages over previous works on ViTs.

[1] Multi-level Logit Distillation, https://openaccess.thecvf.com/content/CVPR2023/papers/Jin_Multi-Level_Logit_Distillation_CVPR_2023_paper.pdf

**Questions:**

See weakness

---

> ### Author Response · Authors · 2023-11-23
>
> Thank you for your feedback. We're glad you found LumiNet's modification to logit Knowledge Distillation simple and effective.
>
> # Comparing with Jin et al. "Multi-Level Logit distillation"
>
> We appreciate your observations. As suggested, we have performed new experiments to compare with Jin et al. work. Please note that Jin et al. considered the Auto-Augmentation technique [1] to benchmark results. In contrast, the benchmarking in our paper did not consider Auto-Augmentation (similar to DKD [2]). Since Jin et al. did not report results without Auto-Augmentation, we exclude comparing our work with them. To ensure a fair comparison, we present updated benchmarking results where Auto-Augmentation has been applied to the CIFAR-100 dataset.
>
> ## Table A. Comparison between Jin et al. and our method after applying Auto Augmentation on CIFAR-100.
>
> | Teacher                  | WRN_40_2  | WRN_40_2  | VGG 13    | ResNet32×4 | WRN-40-2  |
> |--------------------------|-----------|-----------|-----------|------------|-----------|
> | **Accuracy**                | 75.61     | 75.61     | 74.64     | 79.42      | 75.61     |
> | **Student**              | WRN_16_2  | WRN_40_1  | VGG 8     | ShuffleNet-V2 | ShuffleNet-V1 |
> | **Accuracy**                | 73.26     | 71.98     | 70.36     | 71.82      | 70.50     |
> | **Multi-Level**          | 76.63     | 75.35     | 75.18     | 78.44      | 77.44     |
> | **Ours\***               | **76.91** | **76.01** | **75.57** | **79.12**  | **77.97** |
> | **Δ (Ours - Multi-Level)** | +0.28     | +0.66     | +0.39     | +0.68      | +0.53     |
>
> Our method demonstrates significant improvements in performance across five architectures, especially in heterogeneous and homogenous settings, when we apply settings of multi-level logit distillation. DKD [2] further supports such improvement while using auto-augmentation. These results are added in the revised paper.
>
> # Results on ViT
>
> Thank you for your insightful review and for expressing interest in the potential application of our LumiNet method on Vision Transformers (ViTs). We appreciate the suggestion to explore and demonstrate the advantages of LumiNet in the context of ViTs, in addition to convolution-based methods.
>
> Initially, our experiments were focused on convolution-based methods, primarily because the benchmark we utilized is widely recognized and accepted in the research community. Moreover, none of the compared methods in our study had been applied to ViTs, which influenced our initial experimental design. However, we acknowledge the importance and relevance of extending our evaluation to include ViTs. We concur that demonstrating LumiNet's effectiveness on ViTs, similar to its performance on convolution-based networks, would significantly strengthen its applicability and acceptance in diverse scenarios.
>
> Our study evaluated LumiNet with traditional knowledge distillation technique using the CIFAR-100 dataset, operating within a ViT setup. Here, the teacher network is a variant of ViT with 21.3 million parameters, and the student model is a more compact version with approximately 2.38 million parameters. Our findings were quite promising. The improvement margin observed with LumiNet was substantial, as shown in the table below. This suggests that LumiNet excels in convolution-based architectures and holds considerable potential for enhancing performance in ViT. We have used the setting followed by Wang, Jiahao, et al. [3]. These results are added in the revised paper.
>
> ## Table B. Performance on the CIFAR Dataset Using ViT
>
> | Algorithm             | FLOPS  | #params | CIFAR-100 |
> |-----------------------|--------|---------|-----------|
> | Teacher               | ~1.38G | ~21.3M  | 76.30%    |
> | Student               | ~153M  | ~2.38M  | 65.46%    |
> | Knowledge Distillation| ~153M  | ~2.38M  | 67.38%    |
> | Ours                  |  ~153M | ~2.38M  | 68.42%    |
>
> We hope this response addresses your query and demonstrates our commitment to thoroughly exploring and validating the efficacy of LumiNet across various neural network architectures.
>
>
> **References**
> 1. Cubuk, Ekin D., et al. "Autoaugment: Learning augmentation strategies from data." CVPR 2019.
> 2. Zhao, Borui, et al. "Decoupled knowledge distillation." Proceedings of the CVPR 2022.
> 3. Yim, Junho, et al. "A gift from knowledge distillation: Fast optimization, network minimization and transfer learning." CVPR 2017

---

### Author Response · Authors · 2023-11-23

We would like to extend our sincere gratitude for your insightful and constructive feedback on our manuscript.

We appreciate your recognition of the **clarity and simplicity** of our method, LumiNet, and the **effectiveness of our approach** in logit-based knowledge distillation techniques for various vision tasks. Your acknowledgement of our extensive experiments and the **easy-to-follow write-up** is encouraging.

In our rebuttal and subsequent revisions to the manuscript, we have diligently addressed the highlighted weaknesses. **Recognizing the importance of benchmark comparisons**, we have now **included a more thorough analysis and discussion with strong baseline methods**. Additionally, we have **expanded our research to encompass not just Conv-based methods but also explored the application of LumiNet with Vision Transformer architectures**, broadening the scope of our experiments. Addressing concerns about generalization, **we have enriched our manuscript with deeper insights into LumiNet's performance in heterogeneous architecture settings**. Furthermore, **we have made a concerted effort to detail the distinctions and enhancements of our method over similar existing approaches**, alongside more extensive comparisons with state-of-the-art methods.

Once again, we deeply appreciate the valuable insights and suggestions provided by all reviewers.

---

### Meta-Review · Area_Chair_DEHt · 2023-12-04

**Metareview:**

This paper proposes a new logit-based distillation method to distill a teacher model with only predicted logits. The proposed method, LumiNet, can reconstruct more granular inter-class relationships, enabling the student model to learn a richer breadth of knowledge. The proposed LumiNet is evaluated on CIFAR-100, ImageNet, and MSCOCO, revealing its competitive performance.

**Justification For Why Not Higher Score:**

- The concept is not novel. Previous research has delved into examining inter- and intra-class relationships for knowledge distillation, as indicated by the reviewers. Despite the authors' attempts to emphasize distinctions from DKD  and multi-level knowledge distillation, the fundamental contrast between their proposed method and existing approaches remains ambiguous.

- The experiments are only conducted for the student models with similar-architecture as the one of teacher models. It is unclear how well the proposed method would generalize to heterogeneous architectures.

**Justification For Why Not Lower Score:**

NA

---

### Decision · Program_Chairs · 2024-01-16

Reject